# The Intestinal Mechanisms in the Excretion of Pepsinogen, Amylase and Lipase in Coprofiltrate in Women During Pregnancy and the Postpartum Period

**DOI:** 10.3390/biom15081099

**Published:** 2025-07-29

**Authors:** Elena Kolodkina, Sergey Lytaev

**Affiliations:** Department of Normal Physiology, St. Petersburg State Pediatric Medical University, 194100 Saint Petersburg, Russia; 922-666-2045@mail.ru

**Keywords:** pregnancy, digestive enzyme, coprofiltrate, mother-fetus functional system

## Abstract

**Background**: Enzymes secreted by the digestive glands are excreted from the body with urine, sweat and feces, and they are also removed from the blood due to their participation in the enzymatic provision of the secretion entering the gastrointestinal tract. **Objective**: The aim of this work was to analyze the activity of pepsinogen, amylase and lipase in the coprofiltrate of pregnant women in each trimester of pregnancy and in the postpartum period, taking into account the timing and type of delivery (term, premature, late delivery or cesarean section). **Methods**: Data from studies of non-pregnant (n = 45) and pregnant (n = 193) women were analyzed. The materials for preparation coprofiltrate were collected during delivery. Pepsinogen activity was determined by proteolytic activity at pH = 1.5–2.0 using the tyrosine spectrophotometric method, while amylase activity was determined by the amyloclastic method of Karavey, and lipolytic activity was determined by a unified kinetic method using olive oil as a substrate. **Outcomes**: A small amount of pepsinogen was excreted in the coprofiltrate, and while the level of its excretion increased after childbirth, it remained below the control values. At the same time, an increase in the amylolytic activity of the coprofiltrate was observed in all groups of pregnant women examined from the first to the third trimester of pregnancy. In pregnant women, multidirectional changes in lipase activity were observed depending on the timing and type of delivery. **Conclusions**: At the end of pregnancy, amylolytic activity increased in all women, and pepsinase activity decreased compared to the indicators of non-pregnant women. No reliable differences were found in the lipolytic activity of the coprofiltrate in pregnant women at the end of pregnancy and the indicators of non-pregnant women.

## 1. Introduction

Hydrolases enter the digestive tract from the blood, which are then secreted into the intestine and participate in the processing of digestive substrates through step-by-step cavity, peri-mucosal and membrane digestion [1,2]. The restoration of digestive enzymes in the gastrointestinal tract is clearly expressed. Experimental studies on animals, in particular on dogs, have shown that isolated intestinal loops for 20–25% of the digestive capacity are achieved without the participation of pancreatic or prepancreatic digestive enzymes [3,4].

The mechanisms of incretion of digestive gland enzymes have been studied previously [4,5]. The problem of the biological significance of blood hydrolase homeostasis has been partially resolved, and their diverse role in the body has been revealed, including the anabolic, regulatory, informational, transport and other functions of pepsinogen, amylase, and lipase [6,7,8]. According to the data obtained, the concentration of serum amylase reflects the balance between the rates of amylase entry into the blood and its removal from it. Hyperamylasemia can result from either an increased rate of amylase entry into the bloodstream or a decreased metabolic clearance of this enzyme [9,10,11,12].

Physiological changes associated with pregnancy are the result of hormonal and metabolic changes necessary to support the growing fetus. In addition, they lead to changes in the functioning of many body systems, including the gastrointestinal tract [13]. In particular, the effect of pregnancy on irritable bowel syndrome has been shown alongside the efficacy and safety profiles of widely used diets and drugs for this syndrome during pregnancy [14].

During pregnancy, bilateral metabolic relationships are formed between the mother’s organism and the developing fetus [15,16]. The fetus absorbs some of the nutrients by swallowing amniotic fluid, which are enzymatically hydrolyzed to monomers in the gastrointestinal tract of the developing organism. In this regard, digestive hydrolases are of particular importance in the functional system “maternal body—placenta, amniotic fluid—fetus” [17,18,19].

Pregnancy affects homeostasis mechanisms involving the intestine, which is associated with disturbances in the mother–fetus system [20,21,22,23,24]. Secreted enzymes participate in cavity and parietal hydrolysis [4,5,6]. Participation in this digestion of exocrine enzymes, which have not been degraded after their secretion in the overlying sections of the gastrointestinal tract, is possible [25]. In the small intestine, under normal conditions, chyme is formed [25], which is a nutrient medium for eubiotic symbionts of the large intestine, a violation of the composition of which leads to dysbiosis [26]. Along with symbiotic digestion with the participation of enzymes of the bacterial flora, the further hydrolysis of undigested products occurs here by enzymes of resecretory and exocrine origin [27,28]. In the absence of a microvillous apparatus in the large intestine, bacterial cells serve as structures that absorb enzymes [29].

A number of authors have studied changes in the composition of the intestinal microbiome that occur between the first and third trimesters of pregnancy. In particular, an increase in the number of Akkermansia, Bifidobacterium and Firmicutes has been noted, which is associated with an increase in the need for energy accumulation. An increase in the levels of Proteobacteria and Actinobacteria has a protective effect on the mother and the fetus [30].

There is a need to monitor gastrointestinal diseases during pregnancy; in this regard, expert recommendations are proposed regarding the clinical management of patients with gastrointestinal and liver diseases associated with pregnancy [20], determining the use of pharmacokinetic variables of drugs during pregnancy through taking into account the enzymatic activity of biological fluids [21].

Thus, unused hydrolases are excreted in the feces during defecation, and their hydrolytic activity can be detected in the coprofiltrate. Based on the above, the present study was aimed at studying the features of the mechanisms of enzymatic activity of pepsinogen, amylase and lipase in the coprofiltrate of pregnant and parturient women during the trimesters of pregnancy and in the postpartum period, depending on the type and timing of delivery (full-term, premature or late delivery, or cesarean section).

## 2. Materials and Methods

This study involved 193 Caucasian, primiparous pregnant women aged 18 to 35 years with a physiological course in the gestation period. They did not have any pathology of the gastrointestinal tract or liver and did not use drugs, alcohol, or nicotine. To select patients, a clinical interview and preliminary questionnaire of pregnant women were conducted to exclude pathologies of organs and systems. The first group consisted of 161 pregnant women whose births ended through the natural birth canal (96 with term births, 34 with premature births, and 31 with late births—more than 40 weeks). The second group consisted of 32 pregnant women whose births ended in emergency surgery (cesarean section). For select patients, a clinical interview and preliminary questionnaire of pregnant women were performed to exclude pathology of the digestive system and liver. Screening for hepatitis viruses is mandatory in accordance with the national standard. During the study period, no participants reported any complaints of diarrhea or constipation. These data were constantly clarified during the interview with the participants.

The gestational age at the time of delivery was determined by the date of last menstruation, the date of the first fetal movement, the first visit to the antenatal clinic, and data from external obstetric and ultrasound examinations.

The clinical data on the course of pregnancy and childbirth were obtained based on the exchange and notification card for the observation of the pregnant woman and the woman in labor, the history of childbirth, and the history of the development of the newborn.

The control group consisted of 45 practically healthy non-pregnant women aged 18 to 30 years (average 24.2 ± 0.3 years) without any pathologies or concomitant diseases.

This study on women in the St. Petersburg Snegirev’s Maternity Hospital was performed. All women (the control and research groups) were informed about the purpose and methods and gave written voluntary informed consent to participate in the study (protocol no. 0608-23 dated 7 August 2023, of the Local Ethics Committee of the Almazov National Medical Research Center).

The research design consisted of the following stages: coprofiltrate was tested once in healthy subjects (women) and four times in pregnant women regardless of the type of delivery in each trimester of pregnancy and after the birth of the child. To prepare the coprofiltrate, 1 g of the feces being examined was mixed with 10 mL of distilled water. The resulting mixture was then passed through a paper filter. The coprofiltrate was collected for enzyme content determination at the same time of day under the same conditions (in the morning, on an empty stomach). The coprofiltrate was collected once from healthy subjects (women) and four times from pregnant women in each trimester of pregnancy and after childbirth. Early coprofiltrate is a stool sample used to diagnose various conditions related to the gut in the early stages of a disease or condition. Such studies may include an analysis of gut microflora, enzymatic activity, and other parameters. Coprofiltrate collection for enzyme content determination was performed at the same time of day under the same conditions (morning, fasting). Coprofiltrate was collected once from healthy subjects (women) and four times from pregnant women, regardless of the type of disease, in each trimester of pregnancy and after childbirth.

The activities of pepsinogen, amylase and lipase were determined in the coprofiltrate of pregnant women at 12–13, 25–26, and 39–40 weeks of pregnancy and once in individuals in the control group.

Pepsinogen activity was determined by proteolytic activity at pH = 1.5–2.0 by the tyrosine spectrophotometric method. The method is based on the fact that tyrosine, which is formed during the hydrolysis of pepsinogen, absorbs ultraviolet light at a wavelength of about 280 nm. This allows one to determine the concentration of tyrosine and, accordingly, the activity of pepsinogen. The principle of the method is that the protein substrate is incubated with the test liquid (coprofiltrate). The measure of its activity is the amount of tyrosine formed as a result of protein hydrolysis. In this study, lyophilized blood plasma was used as a protein substrate.

Amylase activity was determined using the Karavey amyloclastic method [2]. Experimental and control samples were taken for analysis. A starch solution was added to the experimental sample, which was followed by heating for 5 min in a water bath at 37 °C and adding 0.02 mL of non-hemolyzed serum. The sample was again placed in a water bath for 5 min. The control sample was carried out in the same way but without serum. After that, 1 mL of the working iodine solution and 8 mL of distilled water were added to all test tubes (experimental and control). The solutions were colorimetrically measured on a photometer with a red filter (630–690 nm) in a cuvette with a layer width of 10 mm. Distilled water was used as a standard. A reagent kit for determining the activity of alpha-amylase “Alpha-Amylase-1-Olvex”, Olvex Diagnosticum LLC (Saint-Petersburg, Russia) was used.

Lipolytic activity was determined by a unified kinetic method using olive oil as a substrate, where the unit of enzymatic activity is the amount of enzyme that releases 1 μmol of oleic acid from a 40% olive oil emulsion at pH 7.0 and 37 °C for 1 h. The principle of the method is to spectrophotometrically measure the turbidity of an olive oil suspension under the action of an enzyme. A reagent kit for the quantitative determination of lipase activity by the kinetic method “Lipase-UTS”, Eiliton LLC (Dubna, Russia) was used.

All methods used to study enzymatic activity are approved by national standards and comply with GLP standards.

### Statistical Analysis

The enzyme levels in non-pregnant women were taken as controls to show the differences from those in pregnant women. First, the distribution was checked for normality using the Pearson criterion. The average values (M) and their standard error (m) and standard deviation (σ) were calculated. The dependence between the features was assessed using the pair correlation coefficient (r), its error (mr), and the level of significance of differences (according to Student’s t-criterion). The dependence was considered strong when |r| > 0.7 (average) if the modulus of the pair correlation value was within 0.3–0.7. A correlation value of less than the modal value of 0.3 was considered weak.

The non-parametric Mann–Whitney criterion was used when a non-normal distribution was confirmed. Differences between groups in the levels of the studied characteristics were estimated with the non-parametric Mann–Whitney method using the statistical package SPSS 11.0. Differences were considered significant at an error probability of *p* < 0.05. Significance levels of *p* < 0.01 and *p* < 0.001 were also distinguished. The results are presented as Me [Q1; Q3], where Me is the median, while Q1 and Q3 are the lower and upper quartiles, respectively. The results were statistically processed using Microsoft Excel 2003 spreadsheets and the SPSS 23.0 and Primer of Biostatistics 4.03 programs.

## 3. Results

We have found that out of 82 pregnant women examined during the year, 64 women (78.1%) gave birth through the natural birth canal (42 pregnant women with full-term birth, 12 pregnant women with premature birth, 10 pregnant women with late birth), and 18 women (21.9%) gave birth by emergency surgery—cesarean section.

The average age of pregnant women with natural birth was 30.6 ± 2.1 years (with full-term birth −29.1 ± 1.9 years, premature birth −30.2 ± 3.4 years, late birth −30.8 ± 2.7 years), and in women who gave birth by cesarean section, it was −32.3 ± 5.9 years.

Clinical data in pregnant women with different terms and types of delivery (term delivery (group 1), premature delivery (group 2), late delivery (group 3), delivery by cesarean section (group 4)) are presented in Table 1.

Pepsinogen activity in the coprofiltrate (dilution 1:4) in pregnant women who completed their pregnancy with delivery at term was low and amounted to 410.2 ± 34.1 thousand units/mL. It decreased throughout pregnancy and in the third trimester was ¼ of the initial level −153.8 ± 11.2 thousand units/mL. On the 2nd–3rd day after delivery, the pepsinase activity of the coprofiltrate increased by 2.1 times (*p* < 0.001) compared to the end of pregnancy, but it remained lower −315.3 ± 23.6 thousand units/mL (*p* < 0.05) compared to the control group (Figure 1).

In pregnant women with premature births, there was a decrease in the proteolytic activity of the coprofiltrate throughout pregnancy and in the postpartum period compared to the control values. In the first trimester of pregnancy, pepsinogen activity was 3.5 times lower (*p* < 0.001) than that in non-pregnant women, while it was 1.1 times lower in the second trimester, 2.4 times lower in the third (*p* < 0.001), and 1.2 times lower after childbirth(*p* < 0.05).

Different dynamics were observed in relation to pepsinogen in the coprofiltrate of pregnant women with late labor. There was a decrease in the enzyme activity in the first trimester of pregnancy −172.0 ± 11.3 tyr. U/mL (*p* < 0.001) compared with the control values, while there was an increase in the second trimester −493.1 ± 24.8 tyr. U/mL and decreases in the third trimester (122.4 ± 10.3 tyr. U/mL (*p* < 0.05) and in the postpartum period (263.8 ± 13.7 tyr. U/mL (*p* < 0.001). Such losses of pepsinogen may indicate a greater use of the enzyme in the anabolism of substances in the mother–fetus system, and it may also reflect a decrease in incretion during pregnancy.

Pepsinogen excretion in feces of pregnant women with cesarean section in the first trimester was almost equal to the values of non-pregnant women and amounted to 430.4 ± 32.7 tyr. U/mL (in the control 422.2 ± 28.4 tyr. U/mL). In the second and third trimesters, the pepsinase activity of the coprofiltrate decreased, with the greatest changes at the end of pregnancy −110.3 ± 11.8 tyr. U/mL (*p* < 0.001), remaining low in the postpartum period (192.9 ± 10.6 tyr. U/mL; *p* < 0.001).

In the postpartum period, pregnant women with urgent, premature, late labor, and cesarean section delivery showed a tendency toward increased intestinal enzyme secretion, not reaching the 100% levels seen in non-pregnant women. Thus, a small amount of pepsinogen is secreted in the coprofiltrate, and only after delivery does its secretion level increase, but it remains below control levels. In the postpartum period, intestinal enzyme secretion is maintained for use in subsequent interactions between the nursing mother and her child.

The amylolytic activity of the coprofiltrate in the group of pregnant women who gave birth at term increased by more than 50% in the second and by 127% in the third trimester, decreasing to the initial values after delivery (Figure 2). In pregnancies ending in premature delivery, similar dynamics of changes in the amylolytic activity of the coprofiltrate are observed, as in the case of term delivery, with a slight difference: a reduced enzyme content in the coprofiltrate.

In pregnant women with late labor, the excretion of amylase with feces exceeded the coprofiltrate indices of non-pregnant women. In the first and second trimesters of pregnancy and in the postpartum period, the enzyme activity in the coprofiltrate of the studied group of women was the highest compared to other groups.

In women with cesarean section, the amylolytic activity of the coprofiltrate increased by trimester of pregnancy. In the first trimester, it was 118.9% that of the control, while in the second trimester, it was 177.9%, and in the third trimester, it was 291.2%. In the postpartum period, it sharply decreased, amounting to 74.3%, which is lower than the indicators of non-pregnant women. Thus, an increase in the amylolytic activity of the coprofiltrate was observed in all groups of pregnant women examined, grouped by the timing and types of delivery, from the first to the third trimester of pregnancy.

Lipase excretion by the intestines differed significantly in different groups of women studied. In pregnant women who gave birth at term, the enzyme activity in the coprofiltrate had the highest values in the first trimester (420.3 ± 22.5 U/mL; (*p* < 0.001), which is 31% higher than in non-pregnant women (Figure 3). Subsequently, during pregnancy, it decreased and after delivery was the same as in the control.

Similar dynamics were observed in women with premature delivery (maximum lipase values in the coprofiltrate were in the first trimester, while the lowest were in the postpartum period). A different pattern of changes in the lipolytic activity of the coprofiltrate was observed in women with late delivery. Being reduced in the first trimester of pregnancy compared to the control group data, the excretion of the enzyme in the coprofiltrate increased in the second (105.9%) and third (116.9%) trimesters, reaching maximum values after delivery (131.6%). In women who delivered by cesarean section, during pregnancy and after delivery, differently directed dynamics of changes in intestinal lipase excretion were observed. In the first trimester of pregnancy, the enzyme activity in the coprofiltrate was practically no different from the control group values and was 96.6% of the control. In the second trimester, the lipolytic activity of the coprofiltrate decreased and was 12.7% lower than in non-pregnant women. In the third trimester of pregnancy and after childbirth, intestinal lipase excretion increased, exceeding the control group values by 12.4% and 22.8%.

## 4. Discussion

Thus, pregnancy affects homeostasis mechanisms involving the intestine, which is associated with hormonal and metabolic shifts in the “mother–fetus” system as well as with disruption of the biorhythms of bowel movements. Various mechanisms participate in ensuring the homeostasis of pepsinogen, amylase and lipase, among which is the excretion of hydrolases of the digestive glands [3,5]. Pepsinogen, amylase, and lipase enter the digestive tract; then, these enzymes are resected into the intestine and participate in the complexly organized digestion of substrates through step-by-step cavity, primucosal and membrane digestion [2,4,5].

Pepsinogen is expressed in the main cells of the stomach; the fundic mucous cells and pyloric glandular cells of the gastric epithelium; the mammary gland, prostate gland, lungs and seminal vesicles; and changes in pathological conditions of the body as well as during pregnancy [31].

The results of the present study show that women with term delivery during pregnancy have low pepsinogen activity compared to the control group. This fact toward the end of pregnancy can be associated with the expenditure or consumption of enzymes on anabolic processes of the growing fetus. After birth, the enzyme level increased, but the activity remained lower than in non-pregnant women. The same dynamics were observed in women with premature births, while in late births, there were multidirectional changes in the enzyme activity in the coprofiltrate during pregnancy and in the postpartum period. In women with cesarean section in the first trimester of pregnancy, there were no differences in enzyme activity from the control group, in the second, third trimester and after birth, there was a decrease in the proteolytic activity of the coprofiltrate.

The amylolytic activity of blood serum is normally realized by almost half by the action of pancreatic α-amylase [5]. As is known, amylases have a number of phenotypes that differ in their physicochemical properties, and they also have different renal clearance and an unequal half-life from the body [32]. In particular, there is information on the genetic characteristics of the ratio of β- and s-amylases in human blood serum [33]. Amylases in the blood are free and bound to plasma proteins (serum) and formed elements. At the same time, plasma proteins (albumins and all globulin fractions) have independent amylolytic activity. This is one of the forms of enzyme deposition: to maintain homeostasis of the body’s enzymes by increasing or decreasing the affinity for them depending on hypo- or hyperamylasemia [34].

The amylolytic activity of coprofiltrate in pregnant women with term births and cesarean section increased from the beginning to the end of pregnancy and decreased in the postpartum period. In case of premature birth, the enzyme activity in the coprofiltrate in the first trimester of pregnancy did not differ from the control group indicators, and in the second and third trimesters as well as after birth, it was higher than the control group indicators. In case of late birth, there was an increase in amylase activity in the coprofiltrate throughout pregnancy and in the postpartum period.

The origin of lipase in blood serum is mainly pancreatic and hepatic. It is also found in formed elements of blood (erythrocytes, leukocytes), but serum and erythrocyte lipolytic activities are independent of each other [35].

In the small intestine, membrane digestion is carried out on the microvilli of enterocytes coupled with the processes of transport of nutrient monomers into the villus. In the large intestine, the removal of enzymes from the cavity can be carried out by microorganisms—saprophytes participating in symbiotic digestion.

Unused hydrolases are excreted in the feces during defecation, and their hydrolytic activity can be detected in the coprofiltrate prepared from the feces by dilution with saline in a ratio of 1:4.

The lipolytic activity of coprofiltrate in pregnant women in our study with term and premature births increased in the first trimester, with a subsequent decrease in the second and third trimester as well as after birth. The opposite dynamics was noted in pregnant women with late births—the lowest values were in the first trimester, and the highest were after birth. Multidirectional changes in lipase activity were observed in women with cesarean section (no changes in the first trimester compared to the control, a decrease in the second trimester, an increase in the third trimester and after birth).

During pregnancy, metabolic connections are established between the mother’s body and the growing fetus, which actively absorbs amino acids for protein synthesis. The fetus absorbs nutrients with amniotic fluid, which are hydrolyzed to monomers in the gastrointestinal tract of the developing organism by enzymes secreted into the aquafetal environment during autolytic digestion [35,36,37].

As a result of our studies, the excretory origin of the hydrolytic activity of the coprofiltrate was revealed due to the detection of amylase, pepsinogen and lipase in feces. At the end of pregnancy, amylolytic activity increased in all women, while pepsinase activity decreased by 3 times (*p* < 0.001) compared to the indicators of non-pregnant women. No reliable differences were found in the lipolytic activity of the coprofiltrate in pregnant women at the end of pregnancy and the indicators of non-pregnant women.

Thus, as a result of our studies, it was revealed that at the end of pregnancy, amylolytic activity increased in all women, while pepsinase activity decreased compared to the indices of non-pregnant women. No reliable differences were found in the lipolytic activity of coprofiltrate in pregnant women at the end of pregnancy and the indices of non-pregnant women. During pregnancy, neuro-endocrine changes occur in the female body, while the important role belongs to hormonal shifts. There are intense metabolic processes between the mother’s body, placenta and the fetus. The placenta begins to produce hormones. Changes in the functioning of the liver and pancreas are noted. All these factors affect the change in enzymes during pregnancy in women.

## 5. Limitations

The study included pregnant women with a normal gestation period, primiparous women, and those without gastrointestinal tract pathology. Multiparous women were not taken into account. This requires further research with a wider range of pregnant women. The study examines the activity of digestive enzymes secreted by the intestine during pregnancy and after childbirth. The level of enzymes in other biological fluids (gastric juice) was not assessed. This requires further research with a wider range of assessment of the gastrointestinal tract in pregnant women and nursing mothers. All subjects underwent diagnostic procedures in accordance with current standards—blood tests, necessary functional diagnostics, etc. However, this information is not provided in this paper, since it is not the purpose of this study; instead, they may be presented later.

## 6. Conclusions

The results obtained in the present study point to a number of mechanisms of hydrolase homeostasis: excretion from the blood by the intestine, the action of digestive juices, and the restoration and removal of enzymes. In particular, a small amount of pepsinogen is eliminated in the feces; after delivery, its level of elimination increases but remains below control values. In the postpartum period, pepsinogen homeostasis in the intestine is maintained for its use in the relationship between mother and child care.

The amylolytic activity of coprofiltrate increases in all groups of pregnant women, according to the timing and type of delivery, from the first to the third trimester of pregnancy. Alternatively, this task can be saved for future studies—studying coprofiltrate amylase in pregnant women without gestational diabetes in comparison with pregnant women with GDM. These data may provide a non-invasive way to screen people at risk of gestational diabetes.

For intestinal lipase removal, the heterogeneous dynamics of changes in the indices of women who gave birth by cesarean section during pregnancy and after childbirth are noted. In the first trimester of pregnancy, lipase activity is practically no different from the control group. In the second trimester, the lipolytic activity of the coprofiltrate decreases compared to non-pregnant women. In the third trimester of pregnancy and after childbirth, intestinal lipase excretion increases, significantly exceeding the indices of the control group.

## Figures and Tables

**Figure 1 biomolecules-15-01099-f001:**
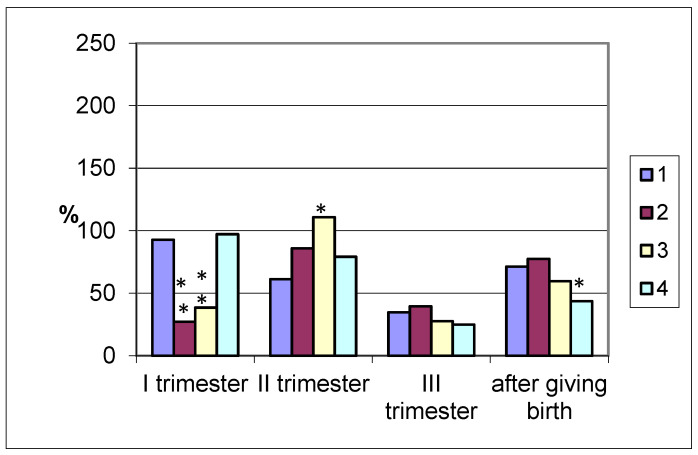
Pepsinogen activity in the coprofiltrate of pregnant women (by trimester of pregnancy and after delivery) with different timing and types of delivery depending on the timing and type of delivery (in % of the control group values taken as 100%). Note. 1—term delivery; 2—premature delivery; 3—late delivery; 4—delivery by cesarean section; significance of differences with values in pregnant women who delivered at term: *—*p* < 0.05; **—*p* < 0.001.

**Figure 2 biomolecules-15-01099-f002:**
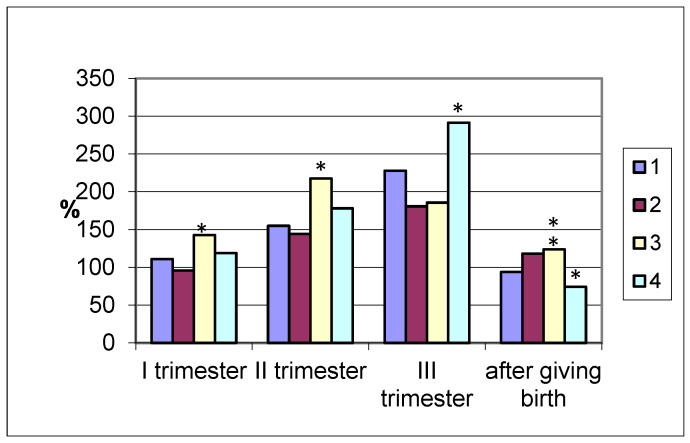
Coprofiltrate amylase activity in pregnant women (by trimester of pregnancy and after delivery) with different timing and types of delivery depending on the timing and type of delivery (in % of the control group values taken as 100%). Note: 1—term delivery; 2—premature delivery; 3—late delivery; 4—delivery by cesarean section; significance of differences with values in pregnant women who delivered at term: *—*p* < 0.05; **—*p* < 0.001.

**Figure 3 biomolecules-15-01099-f003:**
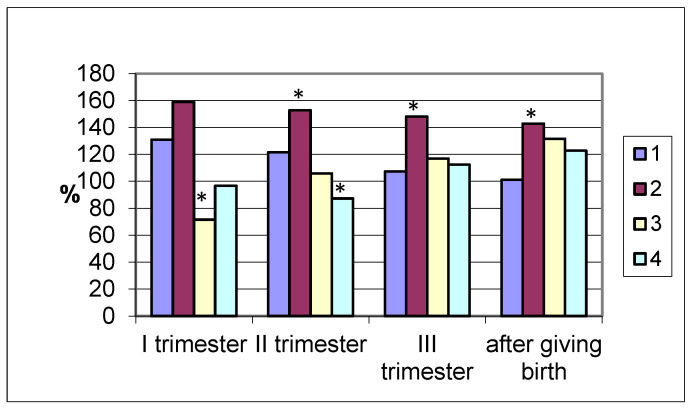
Coprofiltrate lipase activity in pregnant women (by trimester of pregnancy and after delivery) with different timing and types of delivery depending on the timing and type of delivery (in % of the control group values taken as 100%). Note: 1—term delivery; 2—premature delivery; 3—late delivery; 4—delivery by cesarean section; significance of differences with values in pregnant women who delivered at term: *—*p* < 0.05.

**Table 1 biomolecules-15-01099-t001:** Comparative characteristics of clinical indicators in pregnant women with natural delivery and by caesarean section, *—*p* < 0.05, **—*p* < 0.001.

Parameters	Groups of Pregnant Women			
	Timely Birth (n = 86)	Premature Birth (n = 34)	Late Birth (n = 31)	Childbirth by Cesarean Section (n = 42)
Pre-gestational body weight, kg	62.4 ± 1.2	53.5 ± 1.6 *	64.8 ± 1.4 **	60.1 ± 2.2
Height, cm	164.2 ± 12.5	164.6 ± 14.8	168.5 ± 12.7	166.8 ± 13.2
Total weight gain during pregnancy, kg	9.3 ± 0.2	10.3 ± 0.8	12.6 ± 0.7 *	11.8 ± 0.9 **
Pre-gestational body mass index, kg/m^2^	22.0 ± 1.4	20.4 ± 1.8 **	24.5 ± 2.1 **	22.8 ± 2.0
Early toxocosis, %	2.8	2.2	2.5	3.1
Anemia, %	1.5	1.6	1.3	1.4
Weakness of labor activity, %	4.3	3.7	4.8	5.3

## Data Availability

The data for this project are confidential but may be obtained under the data use agreements of the Saint Petersburg State Pediatric Medical University, the Head of the Department of Normal Physiology. Researchers interested in accessing the data may contact Sergey Lytaev at physiology@gpmu.org. It could take some weeks (months) to negotiate the data use agreements and gain access to the data. The author will assist with any reasonable replication attempts for one year following publication.

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
