# Peer review of "The Intestinal Mechanisms in the Excretion of Pepsinogen, Amylase and Lipase in Coprofiltrate in Women During Pregnancy and the Postpartum Period"

_biomolecules, 2025, doi:10.3390/biom15081099_

Round 1
Reviewer 1 Report
Comments and Suggestions for Authors
Dear Authors,
The manuscript submitted for review is interesting and quite well-written. The research experiment was adequately organized and conducted. However, the conclusions expressed in the Discussion do not appear to be sufficient for generalization. Please consider the revision with the following points:
Title
Please clearly indicate that the sample used is feces. It is preferable to change "in the Excretion" to "in the Fecal Excretion" or "in the Coprofiltrate."
Abstract
L13-14 The expression "Coprofiltrate were collected during labor" is not appropriate. It should be the faces that were obtained from labor. Please state "collection of faces" and "preparation of excretion" separately.
Materials and methods
As above, please separate "collection of faces" and "preparation of excretion". First of all, I can't find any mention of "preparation of excretion" anywhere. What procedure was used to prepare coprofiltrate?
Discussion
L317-L319 The conclusions presented here seem insufficient to generalize. Is this based on the results of this study?
Author Response
Dear Reviewer,
First of all, we would like to thank you for your work in reviewing our paper. We fully agree with your comments and have taken them into account when revising the manuscript.
Q: Title Please clearly indicate that the sample used is feces. It is preferable to change "in the Excretion" to "in the Fecal Excretion" or "in the Coprofiltrate."
A: Changed - Mechanisms of Excretion of Pepsinogen, Amylase and Lipase in Coprofiltrate in Women during Pregnancy and in the Postpartum Period
Q: Abstract
A: Changed. Please see the new version of the paper.
Q: L13-14 The expression "Coprofiltrate were collected during labor" is not appropriate. It should be the faces that were obtained from labor. Please state "collection of faces" and "preparation of excretion" separately.
A: Changed in the abstract. Coprofiltrate has been replaced with material.
Q: Materials and methods
As above, please separate "collection of faces" and "preparation of excretion". First of all, I can't find any mention of "preparation of excretion" anywhere. What procedure was used to prepare coprofiltrate?
A: Added The research design consisted of the following stages: coprofiltrate was tested once in healthy subjects (women) and four times in pregnant women, regardless of the type of delivery in each trimester of pregnancy and after the birth of the child. To prepare the coprofiltrate, 1 g of the feces being examined was mixed with 10 ml of distilled water. The resulting mixture was then passed through a paper filter. The coprofiltrate was collected for enzyme content determination at the same time of day under the same conditions (in the morning, on an empty stomach). The coprofiltrate was collected once from healthy subjects (women) and four times from pregnant women in each trimester of pregnancy and after childbirth. Early coprofiltrate is a stool sample used to diagnose various conditions related to the gut in the early stages of a disease or condition. Such studies may include analysis of gut microflora, enzymatic activity, and other parameters. Coprofiltrate collection for enzyme content determination was performed at the same time of day under the same conditions (morning, fasting). Coprofiltrate was collected once from healthy subjects (women) and four times from pregnant women, regardless of the type of disease, in each trimester of pregnancy and after childbirth. The activities of pepsinogen, amylase and lipase were determined in the coprofiltrate of pregnant women at 12–13, 25–26, and 39–40 weeks of pregnancy and once in individuals in the control group.
Q: Discussion L317-L319 The conclusions presented here seem insufficient to generalize. Is this based on the results of this study?
A: The conclusion is based on the most important own results.
Reviewer 2 Report
Comments and Suggestions for Authors
This is an interesting paper concerned the intestinal mechanisms in the excretion of pepsinogen, amylase, lipase in women during pregnancy taking into account type and time of delivery. The conception of this work is well. The literature about this problem is lucking. I strongly reccomend this paper for publication after revision.
In the introduction - please check the correctness the citation no. 17
In the method section - I suggest add the information about the place of patients recruitment (department, out-patients clinic? time and criteria of recruitment - all hospitalized patients?, years of recruitment).
Discussion and conclusion: in the discussion section, I suggest try to explain the results. Why the time and way of delivery may influence on enzyme activity?
Conclusion should be revised – at this moment it's the repetition of the results.
Author Response
Dear reviewer!
Before, we would like to thank you for the work to improve our paper. We completely agree with the comments you proposed and took this into account in the new version of the manuscript.
This is an interesting paper concerned the intestinal mechanisms in the excretion of pepsinogen, amylase, lipase in women during pregnancy taking into account type and time of delivery. The conception of this work is well. The literature about this problem is lucking. I strongly reccomend this paper for publication after revision.
- In the introduction - please check the correctness the citation no. 17
Answer: The source is correct.
- In the method section - I suggest add the information about the place of patients recruitment (department, out-patients clinic? time and criteria of recruitment - all hospitalized patients?, years of recruitment).
Answer: Information added to chapter Methods.
- Discussion and conclusion: in the discussion section, I suggest try to explain the results. Why the time and way of delivery may influence on enzyme activity?
Answer: The discussion is edited. Our main goal was in comparison of our results of assessing the dynamics of Pepsinogen, amylase and lipase with data available in the literature. We conducted this assessment taking into account the physiology of a pregnant woman, as well as other factors that affect enzymatic activity according to other authors. The tasks of future research have been set.
- Conclusion should be revised – at this moment it's the repetition of the results.
Answer: Conclusion was changed.
Reviewer 3 Report
Comments and Suggestions for Authors
This manuscript focuses on evaluation of digestive enzyme activity in stool water of pregnant women compared to healthy non-pregnant women and aims to correlate levels with clinical status.
Introduction:
- It is difficult to critique manuscripts whose authors do not have English as a primary language, but there are several words and sentences in the introduction that do not make sense to me. For example:
- Line 33-34: What is meant by “resected”? Do the authors mean “secreted”?
- Line 35: What is meant by “primucosal”? Do the authors mean “peri-mucosal”?
- Line 35: What is meant by “recretion”? Do the authors mean “secretion”?
- Lines 37-38: The authors say “… 20-25% of the increted enzymes are resected by isolated sections of the small intestine without taking into account their release in the final substrate”. I believe this research shows that in isolated bowel loops, 20-25% of digestive capacity is achieved without the contribution of pancreatic or pre-pancreatic digestive enzymes.
- In general, the Introduction is long and rambling and discusses concepts that seem unrelated to the thrust of the article.
Methods:
- “Late term” needs to be defined
- Lines 168-171 are a repeat of the description of the control population and can be deleted.
- The statistical analysis should be its own paragraph
- A statistician should be consulted about the appropriateness of this analysis and the results – I am not a statistician.
Results:
- It is notable that all of the subjects developed gestational diabetes (lines193-194). This should be a focus in the Discussion.
- Lines 198-221. This text repeats what is shown in Table 1. The Table is clear and is a good way to present the data rather than describing the same information for each group. There is no comment on whether some of these demographic findings are statistically significant. (e.g. the pre-gestational weight and BMI of those who delivered prematurely seems much lower than the other groups; the pre-gestational weight and BMI of those who delivered late term seems higher).
- Figure 1: I am confused. Does “Urgent delivery” mean full-term infants? There is a separate group delivered by cesarian section (what I would term “urgent”). The authors should use the same language with which they describe these groups in the Methods and in Table 1 (Timely birth (which is more appropriately described in the text as “term birth”), Premature birth , Late birth, Childbirth by cesarean section and healthy controls).
- Figure 1, Panel B: the y-axis is different than in the other panels. It should be the same, especially since the value for the non-pregnant healthy control subjects is the same in each graph.
- Lines 243-245: Speculative interpretation of data does not belong in the Results section unless the interpretation is softened by use of words such as “may” or “could”.
- Lines 243-245: Again, a conclusion not based on the data, although the authors can speculate as long as it is clear that it is an interpretation of data and not a fact.
- Lines 252-3: The authors write: “Thus, an insignificant amount of pepsinogen is excreted in feces, only after delivery the level of its excretion increases”. If I look at Figure 1 it appears that pepsinogen levels in coprofiltrate are significantly lower in all groups compared to control subjects in the third trimester and rise but do not achieve levels seen in healthy controls in the early days after delivery.
- Figure 2: Same comments as above regarding the labels of the groups. Panels B and C have different y-axes than panels A and D. The authors should choose one standard.
- Lines 262-3: The authors write: “After delivery, the enzyme activity becomes lower than the initial level and the values of non-pregnant women…” Both levels appear to be within the error bars of the control group and are not significant. This statement should be dropped.
- Lines 264-285: The authors are repeating the data in Figure 2 but have hidden the most important observation in lines 283-285 : amylolytic activity seems to be similar to that of healthy control subjects in the first trimester, but levels are statistically significantly higher for all pregnant groups in the second and third trimesters. Levels then fall within the normal range (I presume this for the Cesarian section group because no p-value is described).
- Lines 286-315: The patterns of lipase activity seem to be somewhat random. To me, what is missing is pointing out the only consistent statistically significant finding, which is that women who delivered prematurely always had higher lipase activity in coprofiltrate than the control group.
- Figure 3, Panel D: The y-axis differs from the other panels.
Discussion:
- Line 329: “Our studies have shown” should be replaced with “Our study shows”. Although the authors performed many analyses, this data is being presented as a whole.
- Lines 329-30: The authors state that “women with term births had low pepsinogen activity compared to the control group”. Based on their data, this was true only in the third trimester. They go on to note this in lines 334-337. This paragraph is the place for the authors to speculate about the meaning of their findings. Why do they speculate that pepsinogen activity decreases towards the end of pregnancy?
- Lines 338-346: I do not know why this information is here. The Methods section indicates that what is being measured is alpha-amylase and therefore this information seems extraneous.
- Lines 347-353: Again, the authors somewhat mis-represent their findings. In addition, this is an opportunity to speculate on what these data mean since all pregnant subjects had gestational diabetes. In the Introduction they note that there is conflicting data about serum amylase in women with GDM. Gestational diabetes typically develops during the second trimester of pregnancy. Is a next step to look at coprofiltrate amylase in pregnant women without GDM compared to those with CDM? Could this data provide a non-invasive way to screen for those at risk of gestational diabetes?
- Lines 354-356: Again, true but not relevant.
- Lines 361-363: This belongs in the Introduction. A better way to make this statement would be: “Unused hydrolases are excreted with feces during defecation, and their hydrolytic activity can be detected in coprofiltrate prepared from feces by diluting 1: 4 with physiological solution”.
- Line 376: I don’t know what is meant by excretory-recretory.
- Lines 382-385: Unlike the title of the manuscript, this study does not provide data about the origin of the enzyme activity detected in coprofiltrate or describe mechanisms of their excretion. In addition, why do the authors mention transaminases here?
- A paragraph on the limitations of this study is missing.
Conclusions:
- This section does not provide a clear summary of the findings or their relevance.
Minor issues:
- Methods: Line 136: “was tested” is two words
- Results: Line 249: The words “In the” should be deleted
- Discussion: Line 317: Delete the word, “Thus”.
- Discussion: Line 359: I think by “sorption” they mean “removal” or “breakdown”
- Bibliography: The citations have double numbers.
It is understandable that authors for whom English is not a primary language have difficulty writing something as complex as a scientific paper. I suggest that the authors find a colleague who is more fluent in English to review their manuscripts prior to submission.
Author Response
Dear reviewer!
First of all, we thank you for your great work on improving this manuscript. We fully agree with your comments. Before giving a step-by-step answer, please take into account the following circumstances that affect the revision and are beyond our control.
- Before your review, the editors began editing the text. Therefore, the text already differs from the original. This is what the editors asked for. The comments on the lines were taken into account as much as possible. Since some of the lines were shifted and some of the text were changed, there may be a discrepancy in the line numbers. We tried to find these phrases in the text.
- Regarding the comments on English. It is difficult to disagree with you. Some changes have been made. If the paper moves towards publication, we will take the service from the publisher to improve the English.
This manuscript focuses on evaluation of digestive enzyme activity in stool water of pregnant women compared to healthy non-pregnant women and aims to correlate levels with clinical status.
Introduction:
- It is difficult to critique manuscripts whose authors do not have English as a primary language, but there are several words and sentences in the introduction that do not make sense to me. For example:
- Line 33-34: What is meant by “resected”? Do the authors mean “secreted”?
Changed to “Secreted”
- Line 35: What is meant by “primucosal”? Do the authors mean “peri-mucosal”?
- Line 35: What is meant by “recretion”? Do the authors mean “secretion”?
Changed to “peri-mucosal”
- Lines 37-38: The authors say “… 20-25% of the increted enzymes are resected by isolated sections of the small intestine without taking into account their release in the final substrate”. I believe this research shows that in isolated bowel loops, 20-25% of digestive capacity is achieved without the contribution of pancreatic or pre-pancreatic digestive enzymes.
- In general, the Introduction is long and rambling and discusses concepts that seem unrelated to the thrust of the article.
Changed: Experimental studies on animals, in particular on dogs, have shown that in isolated intestinal loops 20-25% of the digestive capacity is achieved without the participation of pancreatic or prepancreatic digestive enzymes [3, 4, 5].
Methods:
- “Late term” needs to be defined
Added: "late term" - more than 40 weeks of pregnancy
- Lines 168-171 are a repeat of the description of the control population and can be deleted.
- The statistical analysis should be its own paragraph
- A statistician should be consulted about the appropriateness of this analysis and the results – I am not a statistician.
Answer: The phrase has been removed. Subchapter added – Statistical Analysis.
Results:
- It is notable that all of the subjects developed gestational diabetes (lines193-194). This should be a focus in the Discussion.
Answer: The phrase has been removed.
Answer to Results Chapter: The chapter has been completely changed, along with the table and figures. Unnecessary phrases have been removed, differences have been added where there were none.
- Lines 198-221. This text repeats what is shown in Table 1. The Table is clear and is a good way to present the data rather than describing the same information for each group. There is no comment on whether some of these demographic findings are statistically significant. (e.g. the pre-gestational weight and BMI of those who delivered prematurely seems much lower than the other groups; the pre-gestational weight and BMI of those who delivered late term seems higher).
- Figure 1: I am confused. Does “Urgent delivery” mean full-term infants? There is a separate group delivered by cesarian section (what I would term “urgent”). The authors should use the same language with which they describe these groups in the Methods and in Table 1 (Timely birth (which is more appropriately described in the text as “term birth”), Premature birth , Late birth, Childbirth by cesarean section and healthy controls).
- Figure 1, Panel B: the y-axis is different than in the other panels. It should be the same, especially since the value for the non-pregnant healthy control subjects is the same in each graph.
- Lines 243-245: Speculative interpretation of data does not belong in the Results section unless the interpretation is softened by use of words such as “may” or “could”.
- Lines 243-245: Again, a conclusion not based on the data, although the authors can speculate as long as it is clear that it is an interpretation of data and not a fact.
- Lines 252-3: The authors write: “Thus, an insignificant amount of pepsinogen is excreted in feces, only after delivery the level of its excretion increases”. If I look at Figure 1 it appears that pepsinogen levels in coprofiltrate are significantly lower in all groups compared to control subjects in the third trimester and rise but do not achieve levels seen in healthy controls in the early days after delivery.
- Figure 2: Same comments as above regarding the labels of the groups. Panels B and C have different y-axes than panels A and D. The authors should choose one standard.
- Lines 262-3: The authors write: “After delivery, the enzyme activity becomes lower than the initial level and the values of non-pregnant women…” Both levels appear to be within the error bars of the control group and are not significant. This statement should be dropped.
- Lines 264-285: The authors are repeating the data in Figure 2 but have hidden the most important observation in lines 283-285 : amylolytic activity seems to be similar to that of healthy control subjects in the first trimester, but levels are statistically significantly higher for all pregnant groups in the second and third trimesters. Levels then fall within the normal range (I presume this for the Cesarian section group because no p-value is described).
- Lines 286-315: The patterns of lipase activity seem to be somewhat random. To me, what is missing is pointing out the only consistent statistically significant finding, which is that women who delivered prematurely always had higher lipase activity in coprofiltrate than the control group.
- Figure 3, Panel D: The y-axis differs from the other panels.
Discussion:
- Line 329: “Our studies have shown” should be replaced with “Our study shows”. Although the authors performed many analyses, this data is being presented as a whole.
Chaged
- Lines 329-30: The authors state that “women with term births had low pepsinogen activity compared to the control group”. Based on their data, this was true only in the third trimester. They go on to note this in lines 334-337. This paragraph is the place for the authors to speculate about the meaning of their findings. Why do they speculate that pepsinogen activity decreases towards the end of pregnancy?
Added: pepsinogen activity decreases towards the end of pregnancy, which is probably due to the expenditure or consumption of the enzyme on the anabolic processes of the growing fetus.
- Lines 338-346: I do not know why this information is here. The Methods section indicates that what is being measured is alpha-amylase and therefore this information seems extraneous.
Information saved, but edited with changes.
- Lines 347-353: Again, the authors somewhat mis-represent their findings. In addition, this is an opportunity to speculate on what these data mean since all pregnant subjects had gestational diabetes. In the Introduction they note that there is conflicting data about serum amylase in women with GDM. Gestational diabetes typically develops during the second trimester of pregnancy. Is a next step to look at coprofiltrate amylase in pregnant women without GDM compared to those with CDM? Could this data provide a non-invasive way to screen for those at risk of gestational diabetes?
Answer: This was an erroneous phrase in the results. The phrase has been removed. Alternatively, this task can be saved for subsequent studies - studying the amylase of the coprofiltrate in pregnant women without GDM in comparison with pregnant women with GDM. These data can become a non-invasive method for screening people at risk of gestational diabetes. Added in conclusion.
- Lines 354-356: Again, true but not relevant.
Information saved.
- Lines 361-363: This belongs in the Introduction. A better way to make this statement would be: “Unused hydrolases are excreted with feces during defecation, and their hydrolytic activity can be detected in coprofiltrate prepared from feces by diluting 1: 4 with physiological solution”.
Thanks. It is saved.
- Line 376: I don’t know what is meant by excretory-recretory.
Saved, as “excretory”
- Lines 382-385: Unlike the title of the manuscript, this study does not provide data about the origin of the enzyme activity detected in coprofiltrate or describe mechanisms of their excretion. In addition, why do the authors mention transaminases here?
Answer: removed transaminases, updated the interim summary of the discussion: Thus, as a result of our studies, it was revealed that at the end of pregnancy, amylolytic activity increased in all women, and pepsinase activity decreased compared to the indicators of non-pregnant women. No reliable differences were found in the lipolytic activity of the coprofiltrate in pregnant women at the end of pregnancy and the indicators of non-pregnant women.
- A paragraph on the limitations of this study is missing.
Added.
Conclusions:
- This section does not provide a clear summary of the findings or their relevance.
Changed.
Minor issues:
- Methods: Line 136: “was tested” is two words
- Results: Line 249: The words “In the” should be deleted
- Discussion: Line 317: Delete the word, “Thus”.
- Discussion: Line 359: I think by “sorption” they mean “removal” or “breakdown”
- Bibliography: The citations have double numbers.
Comments on the Quality of English Language
It is understandable that authors for whom English is not a primary language have difficulty writing something as complex as a scientific paper. I suggest that the authors find a colleague who is more fluent in English to review their manuscripts prior to submission.
Answer: English comments corrected as recommended. If the paper moves to publication, editing service will be ordered.
Duplicate numbering of sources is a result of the publisher's work. Edited in this version.
Reviewer 4 Report
Comments and Suggestions for Authors
- Please describe in details about the sample taking procedures to clear understating of th e readers.
- How many times did the authors take faces samples from the participant? Did the authors determine the time (morning or night) for the sample taking time?
- The authors described that the participants had no GI tract and liver diseases. How did the authors confirm all the participants had no GI tract and liver diseases? Please describe at the revised manuscript. Did the authors screen for hepatitis viruses screening?
- The authors described that sample was taking at each trimester and please describe in details about the specific week for each trimester or not?
- Any participants got any causes of diarrhea or constipation during the study period? How did the author determine for those cases?
- Please describe the study design and sample size determining procedure at the revised manuscript.
- Did the author take written or verbal informed consent? Please describe clearly at the revised manuscript.
Author Response
Dear Reviewer! First of all, we thank you for reviewing our manuscript. We fully agree with your comments and have taken them into account in the revised version of the paper:
Q: Please describe in details about the sample taking procedures to clear understating of th e readers.
A: To prepare the coprofiltrate, 1 g of the feces being examined was mixed with 10 ml of distilled water. The resulting mixture was then passed through a paper filter.
Q: How many times did the authors take faces samples from the participant? Did the authors determine the time (morning or night) for the sample taking time?
A: The coprofiltrate was collected for enzyme content determination at the same time of day under the same conditions (in the morning, on an empty stomach). The coprofiltrate was collected once from healthy subjects (women) and four times from pregnant women in each trimester of pregnancy and after childbirth. The activity of pepsinogen, amylase and lipase was determined in the coprofiltrate of pregnant women at 12-13, 25-26 and 39-40 weeks of pregnancy and once from individuals in the control group.
Q: The authors described that the participants had no GI tract and liver diseases. How did the authors confirm all the participants had no GI tract and liver diseases? Please describe at the revised manuscript. Did the authors screen for hepatitis viruses screening?
A: To select patients, a clinical interview and preliminary questionnaire of pregnant women were conducted to exclude pathology of the digestive system and liver. Screening for hepatitis viruses is mandatory in accordance with the national standard.
Q: The authors described that sample was taking at each trimester and please describe in details about the specific week for each trimester or not?
A: The activity of pepsinogen, amylase and lipase was determined in the coprofiltrate of pregnant women at 12-13, 25-26 and 39-40 weeks of pregnancy and once from individuals in the control group.
Q: Any participants got any causes of diarrhea or constipation during the study period? How did the author determine for those cases?
A: During the study period, no participants reported any complaints of diarrhea or constipation. These data were constantly clarified during the interview with the participants.
Q: Please describe the study design and sample size determining procedure at the revised manuscript.
A: The study involved 193 primiparous women aged 18 to 35 years with a physiological pregnancy. The first group consisted of 161 pregnant women whose birth ended through the natural birth canal (96 full-term, 34 premature and 31 late births). The second group consisted of 32 pregnant women whose birth ended with an emergency operation (caesarean section). The control group consisted of 45 practically healthy non-pregnant women aged 18 to 30 years (average 24.2 ± 0.3 years) without any pathologies or concomitant diseases.
Did the author take written or verbal informed consent? Please describe clearly at the revised manuscript.
The editors separately asked us to send consent forms. We sent the forms approved by the Ministry of Health in Russian and English.
All persons (control and study groups) were familiarized with the purpose and methods, gave written voluntary informed consent to participate in the study
(protocol no. 0608-23 dated 7 August 2023, of the Local Ethics Committee of the Almazov National Medical Research Center, Saint Petersburg).
Reviewer 5 Report
Comments and Suggestions for Authors
The manuscript presents a novel physiological investigation into the excretion patterns of key digestive enzymes pepsinogen, amylase, and lipase in pregnant and postpartum women, with subgroup analyses based on delivery timing and type. The topic is important and relevant, particularly in understanding maternal fetal physiology and enzymatic homeostasis. However, the manuscript requires major revisions before being suitable for publication. The concerns include linguistic clarity, conceptual repetition, data presentation issues, and insufficient mechanistic insight. The following recommendations must be addressed to enhance the manuscript’s quality and scientific rigor.
- Abstract: Make it more concise and highlight key findings with specific data.
- Page 1, Line 33: The term “increted” is non-standard. Likely intended as “incretion,” which refers to hormone secretion into the bloodstream, but usage is unclear. Also clarify or revise the term “resected.”
- Introduction: The literature review is overly long and contains redundancies remove repetitive content.
- Language and Style: The manuscript contains awkward phrasing, grammatical errors, and repeated misuse of non-standard terms like “incretion,” “resecretion,” and “recreation.”
- Methods: Define “coprofiltrate” early, explaining how it differs from standard fecal analysis. Clarify sample collection timing post-delivery.
- Figures and Tables: Figures 1–3 and tables lack adequate description. Move raw data to supplementary tables. Use graphs to show trends (e.g., increase/decrease) and summarize key patterns in text. Ensure clarity in axis labels and captions.
- Discussion: Provide a physiological explanation for enzyme excretion changes during pregnancy, referencing mechanisms such as hormonal shifts, placental clearance, or hepatic/pancreatic changes.
- Conclusion: Avoid speculative claims about “enzyme homeostasis mechanisms” without experimental support. Base conclusions on observed trends only.
The English could be improved to more clearly express the research.
Author Response
Dear reviewer!
Before, we thank you for the work to improve our manuscript. We completely agree with the comments and were completely taken into account as possible during processing. The paper is redone very much. All changes are highlighted by a marker. The chapter is completely redone the results with figures, description and table.
- Abstract: Make it more concise and highlight key findings with specific data.
Answer: Abstract was changed.
- Page 1, Line 33: The term “increted” is non-standard. Likely intended as “incretion,” which refers to hormone secretion into the bloodstream, but usage is unclear. Also clarify or revise the term “resected.”
Answer: Changed on text to «incretion» и «resecretion»
- Introduction: The literature review is overly long and contains redundancies remove repetitive content.
Answer: The review is edited, but the basic information on the enzymes studied is preserved - pepsinogen, amylase and lipase. It is not very appropriate to reduce more.
- Language and Style: The manuscript contains awkward phrasing, grammatical errors, and repeated misuse of non-standard terms like “incretion,” “resecretion,” and “recreation.”
Answer: Many corrections have been made in the text and the adjustment of the terms. If the paper moves in the press, the English editing service will be ordered in the publishing house.
- Methods: Define “coprofiltrate” early, explaining how it differs from standard fecal analysis. Clarify sample collection timing post-delivery.
Answer: Added to Methods
A coprofiltrate at an early stage of development is a model of feces that is used to diagnose various states associated with the intestines at the early stages of the disease or condition. Such studies may include an analysis of intestinal microflora, enzymatic activity and other indicators.
The capture of the coprofiltrate to determine the content of enzymes was carried out at the same time in the same conditions (in the morning, on an empty stomach).
The coprofiltrate was collected once in healthy subjects (women) and four times in pregnant women, regardless of the type of disease, in every trimester of pregnancy and after the birth of a child.
- Figures and Tables: Figures 1–3 and tables lack adequate description. Move raw data to supplementary tables. Use graphs to show trends (e.g., increase/decrease) and summarize key patterns in text. Ensure clarity in axis labels and captions.
Answer: The chapter of the results is completely changed with drawings and a table.
- Discussion: Provide a physiological explanation for enzyme excretion changes during pregnancy, referencing mechanisms such as hormonal shifts, placental clearance, or hepatic/pancreatic changes.
Answer: Added:
During pregnancy, neuro-endocrine changes occur in the female body, while the important role belongs to hormonal shifts. There are intense metabolic processes between the mother’s body, placenta and the fetus. The placenta begins to produce hormones. Changes in the functioning of the liver and pancreas are noted. All these factors affect the change in enzymes during pregnancy in women.
- Conclusion: Avoid speculative claims about “enzyme homeostasis mechanisms” without experimental support. Base conclusions on observed trends only.
Answer: Conclusion was changed.
Round 2
Reviewer 1 Report
Comments and Suggestions for Authors
I have no further questions.
Author Response
Dear Reviewer,
We highly appreciate your work in thoroughly reviewing our paper and are grateful to you.
Respectfully yours
Reviewer 2 Report
Comments and Suggestions for Authors
I accept the corrections.
Author Response

(The authors gave the same response as above.)

Reviewer 3 Report
Comments and Suggestions for Authors
Response to revised manuscript “The Intestinal Mechanisms in the Excretion of Pepsinogen, Amylase and Lipase in Coprofiltrate in Women during Pregnancy and the Postpartum Period”
Introduction:
- The Introduction remains long and rambling and discusses concepts that seem unrelated to the thrust of the article. The last three paragraphs focus on GI aspects of pregnancy but still do not explain why studying the excretion of pepsinogen, amylase and lipase is important.
- The sentence in the Discussion, “Unused hydrolases are excreted in the feces during defecation, and their hydrolytic activity can be detected in the coprofiltrate” belongs in the Introduction so we can understand that levels of hydrolases in coprofiltrate has the potential to be a useful measurement.
Methods:
- This section remains somewhat rambling. For example: “Coprofiltrate was collected once from healthy subjects (women) and four times from pregnant women, regardless of the type of disease, in each trimester of pregnancy and after childbirth” is written twice, followed by a similar sentence with more specific information about the timing of collection of samples.
- Did these women have gestational diabetes or not? How was this determined? This seems very important in light of the exploration of amylase.
Results:
- The authors make a point of studying different length and modes of delivery and list the demographics of each group but never comment on whether they are statistically significant (or not).
- The Figures are improved by using values for the control group as 100%. Drawing a thick line at that 100% level would help the reader since each graph uses a different scale. Rather than listing groups by number, a text key would be better (e.g. “term, early, late, C-S”)
- Are the third trimester values for pepsinogen truly non-significant compared to control values? They are visually similar to the significant values in the first trimester for the premature and late delivery groups.
- Speculative sentences such as “Such losses of pepsinogen may indicate greater use of the enzyme in the anabolism of substances in the mother-fetus system, and may also reflect a decrease in incretion during pregnancy” and the last sentence of the Results section (“Thus, pregnancy affects homeostasis mechanisms…”) belong in the Discussion.
- Figure 2: Are the third trimester values for the late term group truly non-significant compared to control values? They are higher than the significant values in the first trimester.
- Figure 3: Are the first trimester values for the premature delivery group truly non-significant compared to control values? They are higher than the significant values in the subsequent trimesters.
Discussion:
- The Discussion continues to have extraneous information not related to this study, such as the description of the amylolytic activity or lipase in blood.
- The authors do not describe limitations of this study. They have a focus on blood levels in their Introduction and discussion – was it a limitation that blood was not measured?
- I still don’t understand why these findings are important.
Conclusion:
- This section still does not provide a clear summary of the findings or their relevance.
It is understandable that authors for whom English is not a primary language have difficulty writing something as complex as a scientific paper. I suggest that the authors find a colleague who is more fluent in English to review their manuscripts prior to submission.
Author Response
Dear Reviewer!
First of all, we highly appreciate your work on the detailed elaboration of our paper. We have taken into account your new comments as much as possible. The Introduction has been shortened, and unnecessary parts have been removed from the text. Below are our step-by-step answers.
Introduction:
- The Introduction remains long and rambling and discusses concepts that seem unrelated to the thrust of the article. The last three paragraphs focus on GI aspects of pregnancy but still do not explain why studying the excretion of pepsinogen, amylase and lipase is important.
- The sentence in the Discussion, “Unused hydrolases are excreted in the feces during defecation, and their hydrolytic activity can be detected in the coprofiltrate” belongs in the Introduction so we can understand that levels of hydrolases in coprofiltrate has the potential to be a useful measurement.
Answer: The introduction has been shortened and unnecessary phrases have been removed.
Methods:
- This section remains somewhat rambling. For example: “Coprofiltrate was collected once from healthy subjects (women) and four times from pregnant women, regardless of the type of disease, in each trimester of pregnancy and after childbirth” is written twice, followed by a similar sentence with more specific information about the timing of collection of samples.
- Did these women have gestational diabetes or not? How was this determined? This seems very important in light of the exploration of amylase.
Answer: Unnecessary phrases have been removed in accordance with the recommendations. The women examined did not have gestational diabetes, which was determined based on their questionnaire and a biochemical blood test (by glucose level).
Results:
- The authors make a point of studying different length and modes of delivery and list the demographics of each group but never comment on whether they are statistically significant (or not).
Answer: only the average age of pregnant women is described, the rest of the data in the text is not described, since everything is available in the table with an indication of statistically significant differences.
- The Figures are improved by using values for the control group as 100%. Drawing a thick line at that 100% level would help the reader since each graph uses a different scale. Rather than listing groups by number, a text key would be better (e.g. “term, early, late, C-S”)
Answer: We considered it inappropriate to indicate the names of the groups in the legend, since then the Figure itself would be small and the columns would be hard to see, so we did it by numbers. The explanation of the group numbers is below the Figure.
- Are the third trimester values for pepsinogen truly non-significant compared to control values? They are visually similar to the significant values in the first trimester for the premature and late delivery groups.
Answer: the comparison on the graphs is with urgent births. Compared with the control group, significant changes in pepsinogen in the coprofiltrate are noted in the third trimester.
- Speculative sentences such as “Such losses of pepsinogen may indicate greater use of the enzyme in the anabolism of substances in the mother-fetus system, and may also reflect a decrease in incretion during pregnancy” and the last sentence of the Results section (“Thus, pregnancy affects homeostasis mechanisms…”) belong in the Discussion.
Answer: The unnecessary has been removed.
- Figure 2: Are the third trimester values for the late term group truly non-significant compared to control values? They are higher than the significant values in the first trimester.
Answer: The third trimester values for the last group are significant compared to the control values. They are higher than the significant values in the first trimester.
- Figure 3: Are the first trimester values for the premature delivery group truly non-significant compared to control values? They are higher than the significant values in the subsequent trimesters.
Answer: The first trimester values for the preterm birth group are statistically significant compared with the control values.
Discussion:
- The Discussion continues to have extraneous information not related to this study, such as the description of the amylolytic activity or lipase in blood.
- The authors do not describe limitations of this study. They have a focus on blood levels in their Introduction and discussion – was it a limitation that blood was not measured?
- I still don’t understand why these findings are important.
Answer:
Limitations
The study included pregnant women with a normal gestation period, primiparous women, and those without gastrointestinal tract pathology. Multiparous women were not taken into account. This requires further research with a wider range of pregnant women. The study examines the activity of digestive enzymes secreted by the intestine during pregnancy and after childbirth. The level of enzymes in other biological fluids (gastric juice) was not assessed. This requires further research with a wider range of assessment of the gastrointestinal tract in pregnant women and nursing mothers. All subjects underwent diagnostic procedures in accordance with current standards - blood tests, necessary functional diagnostics, etc. However, this information is not provided in this paper, since it is not the purpose of this study and may be presented later.
Conclusion:
- This section still does not provide a clear summary of the findings or their relevance.
Answer: The conclusion is based on the results obtained for the enzymes studied.
Reviewer 4 Report
Comments and Suggestions for Authors
Satisfactory improvements
Author Response

(The authors gave the same response as above.)

Reviewer 5 Report
Comments and Suggestions for Authors
The present manuscript is recommended for publication. The only concern is its high similarity index and the English language.
Comments on the Quality of English LanguageThe English could be improved to more clearly express the research.
Author Response
Dear Reviewer,
We highly appreciate your work in thoroughly reviewing our paper and are grateful to you.
Answer: Unfortunately, we do not see the similarity index. The manuscript is assessed for similarity during the incoming inspection at the editorial office. Usually, the editorial office immediately returns the paper for rephrasing. We did not have this. With regard to English, maximum changes have been made.
Respectfully yours